# Phenological and Fungal Interactions of *Malesherbia auristipulata* Ricardi (Passifloraceae) in the Atacama Desert: Adaptations and Conservation in an Extreme Ecosystem

**DOI:** 10.3390/plants13213035

**Published:** 2024-10-30

**Authors:** German F. Sepúlveda Chavera, Eliana Belmonte Schwarzbaum, Nicolas Valderrama Saez, Mabel Arismendi Macuer, Wilson Huanca-Mamani

**Affiliations:** 1Departamento de Recursos Ambientales, Facultad de Ciencias Agronómicas, Universidad de Tarapacá, Avda. General Velásquez 1775, Arica 100190, Chile; nmvsaez@gmail.com (N.V.S.); arismendimabel@gmail.com (M.A.M.); whuanca@academicos.uta.cl (W.H.-M.); 2Departamento de Biología, Facultad de Ciencias, Universidad de Tarapacá, Avda. General Velásquez 1775, Arica 100190, Chile; eliana.belmonte@gmail.com

**Keywords:** endophytic fungi, biodiversity, endemic vegetation, Chile

## Abstract

*Malesherbia auristipulata* is an endemic plant species of the Atacama Desert, with unique morphological and physiological adaptations. This research was conducted at Cuesta El Águila, Arica and Parinacota Region, Chile. Adult and juvenile plants were monitored, recording their growth, flowering, and fruiting phases. Additionally, plant community species were identified. For the study of endophytic mycoflora, samples of seeds, roots, stems, and leaves were collected, disinfected, and cultivated in specific media. The isolated fungi were analyzed morphologically and molecularly, determining their distribution in different plant organs. The diversity of endophytic fungi associated with *M. auristipulata* and the associated fungal community was determined. The presence of endophytic fungi varied depending on the organ studied, suggesting dynamic interactions in the structure of its fungal community. Among the identified endophytic fungi, *Alternaria sorghi*, *A. alstroemeriae*, and *Fusarium nurragi* stand out for their presence in the root and stem of the plant. Of particular interest is the presence of *F. circinatum* in the leaves. This study provides valuable information for the conservation of *M. auristipulata* and other organisms in the Atacama Desert, highlighting the importance of ecological interactions in the resilience of plants to extreme environmental conditions.

## 1. Introduction

The Atacama Desert is recognized as one of the driest and most extreme environments on the planet, hosting a unique biodiversity that has developed exceptional adaptations to survive under severe environmental conditions [1,2,3]. The aridity of this region is attributed to a combination of geographic and climatic factors, such as the Humboldt Current and the orographic shadow created by the Andes, which block the entry of moisture [4]. This desert serves as an ideal natural laboratory for studying biological adaptation and ecological resilience. The species inhabiting the Atacama Desert have developed extraordinary mechanisms to store water and withstand intense ultraviolet radiation. A prominent example is *Malesherbia auristipulata*, an endemic plant with remarkable morphological and physiological adaptations, such as reduced leaves and thick cuticles that minimize water loss, as well as a deep root system that allows water extraction from the deepest layers of the soil [5,6].

Cuesta El Águila is the only place in Chile where the population of *Malesherbia auristipulata* has been described (Figure 1), extending at an altitude between 1810 and 1834 m above sea level. This area is part of the Sierra de Huaylillas, whose elevations range from 1540 to 2000 m, and was originated during the late Oligocene and early Miocene [7,8]. The region is characterized by extreme aridity, with an average annual precipitation of only 50 mm [9]. Precipitation at higher altitudinal levels is concentrated during the summer months, which increases the relative humidity of the air between late October and March [10]. Additionally, this region is vulnerable to flash floods caused by heavy rains, such as the one that occurred in January–February 2019 [11]. Despite its adaptations to the extreme environment of the Atacama Desert, *M. auristipulata* faces significant threats due to habitat loss and degradation. In Chile, this species was initially classified as rare [12]. It is currently listed as critically endangered [13], considering that the only known population in Chile occupies a very isolated basin and faces significant current and potential threats, including habitat degradation due to anthropogenic factors.

Phenology, which refers to the life cycles and the timing of plant interactions with their environment, is a crucial aspect for understanding the ecology of arid ecosystems. In the Atacama Desert, the phenology of *M. auristipulata* is closely linked to precipitation patterns and water availability. The synchronization of flowering and fruiting with rainfall events is fundamental for the reproductive success of plants in this extreme environment [14]. In this context, filamentous endophytic fungi, which live within plant tissues without causing apparent harm, play an essential role in the phenology and adaptation of plants to adverse environmental conditions [15,16]. Recent studies have shown that these fungi can enhance plant resistance to drought and diseases, directly influencing their survival and reproductive success [17,18]. Additionally, these microorganisms modify nutrient absorption and water use efficiency in plants [19].

The interaction between *M. auristipulata* and endophytic fungi is an emerging area of research that provides valuable insights into the adaptation strategies of plants in extreme environments. It has been demonstrated that endophytic fungi not only improve tolerance to abiotic stress but also influence plant resistance to pathogens [20]. Their presence in *M. auristipulata* is key to explaining its survival capability in one of the driest deserts in the world. However, little is still known about the specific relationship between the phenology of *M. auristipulata* and the diversity of filamentous fungi in the Atacama Desert.

This study presents results on the diversity of filamentous fungi associated with *M. auristipulata* at different phenological stages of the plant. Through field sampling techniques and molecular analysis, the taxonomic composition and relative abundance of these fungi were characterized in different parts of the plant, as well as in the surrounding substrate. Additionally, the correlation between the phenology of *M. auristipulata* and the structure of its associated fungal community was evaluated, with the aim of better understanding plant–fungi interactions in this extreme ecosystem and their relevance for the adaptation and conservation of species in the Atacama Desert.

## 2. Results and Discussion

### 2.1. Surrounding Plant Species in Its Natural Habitat

The distribution of the plant community *M. auristipulata*, classified according to their botanical family and geographic distribution in Chile, is presented in Table 1 [19]. It is important to note that no introduced species were found in the study area, which underscores the ecological integrity of this habitat. All recorded species are xerophytes, adapted to conditions of extreme aridity, and a significant proportion of them are endemic, which reinforces the uniqueness and conservation value of this ecosystem. The species are detailed according to their botanical family, specifying whether they are native or endemic, as well as their geographic distribution across different regions of the country. This information is crucial for understanding the floristic composition and ecological adaptations of the plant community in this arid ecosystem.

The plant community *M. auristipulata* is composed of 18 species, of which 8 are endemic and 10 are native. The distribution of these species by family is as follows: *Aizoaceae* (1 species), *Asteraceae* (4 species), *Chenopodiaceae* (2 species), *Fabaceae* (1 species), *Malvaceae* (2 species), *Montiaceae* (1 species), *Nyctaginaceae* (1 species), *Passifloraceae* (1 species), *Solanaceae* (4 species), and *Verbenaceae* (1 species). It is noteworthy that no introduced species were found in the study area, which is consistent with previous studies [5,20] (Figure 2).

Several plant families were observed, with the Solanaceae family standing out, comprising four species and a total of 723 individuals (674 of *Nolana rhombifolia*, 37 of *Reyesia juniperoides*, 7 of *Solanum peruvianum*, and 5 of *Exodeconus flavus*). This family is clearly dominant in the community. The Asteraceae family is also notable, with three recorded species and a total of 118 individuals (55 of *Ambrosia artemisioides*, 47 of *Senecio zapahuirensis*, and 16 of *Trixis cacalioides*).

The total number of individuals recorded at 49 sampling points along a 20 m linear transect highlights the abundance of different species. A total of 18 species were registered, indicating significant diversity in the studied area. The most abundant species was *Nolana rhombifolia*, with 674 individuals, representing approximately 51% of the total individuals recorded (1,319). *Malesherbia auristipulata* ranked second with 385 individuals, accounting for roughly 29% of the total. Species such as *Tarasa operculata*, *Glandularia gynobasis*, and *Tetragonia microcarpa* had only one individual each, suggesting they are less common in the area. The distribution is uneven, with the two most abundant species comprising over 80% of the total individuals. This suggests that these species either have competitive advantages in the habitat or that the environmental conditions favor their growth. The predominance of a few species could have implications for ecosystem health, as ecosystems with high diversity tend to be more resilient to environmental changes and disturbances. The low abundance of other species may indicate potential issues with local biodiversity, warranting further monitoring or conservation measures.

In environments with harsh conditions, some species can facilitate the growth of others. For instance, species like *Atriplex glaucescens* might create a microhabitat that benefits other plants by providing shade or retaining moisture in the soil. On the other hand, some species may engage in mutualistic relationships. For example, certain plants, like Malesherbia, can interact with fungi or other microorganisms (as discussed later) that help them absorb nutrients in exchange for sugars produced through photosynthesis, or by activating defense mechanisms. These interactions are well documented in the scientific literature, with numerous examples available.

### 2.2. Phenological Study of Malesherbia auristipulata in Its Natural Habitat

The annual phenology of *M. auristipulata* plants in Cuesta El Águila reveals the phenological events throughout the year, such as flowering, fruiting, and other life cycles, suggesting a seasonal phenological trend and its relationship with environmental factors, such as altitude and geographical location [9].

Table 2 presents a monthly phenological record, detailing the growth stages of *M. auristipulata*, documenting the different phenological stages throughout the year, and providing a detailed view of its life cycle in relation to the specific environmental conditions of the region. It allows a comparative analysis of adult and juvenile plants, providing information on their ability to adapt to changing environmental conditions as well as about the vegetation’s resilience when facing natural disasters.

Throughout the year, juvenile plants are predominantly in a vegetative state, especially in the early months, with 100% in the vegetative state in January. As the months progress, a slight increase in flowering and fruiting phases is observed, reaching a peak in August and September (13% in flowering and 8.7% in fruiting). Latency begins to increase in October, reaching 17.4% in November, while the vegetative state gradually decreases. In the case of adult plants, greater phenological variability is observed throughout the year. At the beginning, the majority remain in a vegetative state (68.4% in January), with a small percentage flowering and fruiting. The flowering and fruiting phases increase significantly between June and September, reaching a peak of fruiting in July (47.4%) and flowering in August (26.3%). Toward the last months of the year, latency increases while the percentage of plants in the vegetative and fruiting states decreases.

### 2.3. Endophytic Fungi Associated with Malesherbia auristipulata

The study of the diversity of endophytic fungi associated with *Malesherbia auristipulata* is based on their genetic identity, morphological characteristics of the colonies, taxonomically valuable structures, and their presence in different parts of the plant, such as the root, stem, and leaf. This information is crucial for understanding the possible implications of these fungi on the health of the plant and its ability to adapt to the environment. Additionally, Table 3 presents the identification of endophytic fungi species using the BLASTN bioinformatics tool [21]. This algorithm compares the DNA sequences of the fungi with a database to find the most similar sequence, allowing for species determination. The percentage of identity (% Identity) reflects the similarity between the DNA sequence of the endophytic fungi and the reference sequences stored in databases like NCBI. A higher percentage of identity indicates a greater similarity between the compared sequences [22]. Furthermore, Table 3 indicates the presence of endophytic fungi in different parts of the plant (*M. auristipulata*), such as the root, stem, and leaf. The presence of fungi in a specific sample is indicated by a “+” mark, while the absence of this mark indicates that no fungi were detected in that part of the plant. This detection method is commonly used in plant microbiota studies and has proved to be very efficient in identifying fungal colonization patterns in different plant tissues [16].

The data reveal a community of endophytic fungal species found in multiple samples and in different parts of the plant. Some fungi, such as *Alternaria sorghi*, *A. alstroemeriae*, and *Fusarium nurragi*, exhibit distribution in both the stem and root; *Aureobasidium melanogenum* is found only in the stem, and *Fusarium circinatum* only in the leaf. This pattern could be influenced by factors such as the plant’s genetic variability, soil composition, and environmental conditions [23]. The level of identity between species and isolates is variable. For example, *Alternaria sorghi* NR_160246.1 presents an identity range that varies between 99.13% and 99.83%, indicating a high genetic similarity among different isolates of this species. This phenomenon is consistent with what is expected in population genetics studies, where isolates of the same species usually show a high genetic relationship even in different environments [15]. These authors reported that *Alternaria* species are common in similar environments. Studying plant microbiota is fundamental for understanding plant–fungi interactions and their impact on plant health and adaptation to their environment [17].

The diversity of endophytic fungal species present in the analyzed sample, including genera such as *Alternaria*, *Fusarium*, and *Aureobasidium*, among others, is clearly reflected in Table 3. This taxonomic variability can be considered an indication of the complexity and health of the plant ecosystem studied [17] (Figure 3).

The phenological cycle of *Malesherbia auristipulata*, showing that juvenile plants remain mostly in the vegetative state throughout the year, while adult plants exhibit greater phenological changes, especially during the winter and spring periods, when fruiting and flowering increase (Figure 4).

The diversity in the morphology of the colonies of endophytic fungi obtained from *A. auristipulata*, on APD and 10 days of incubation at 22 ± 2 °C represents the richness of microorganisms associated with plants in desert environments, as manifested in this case (Figure 5).

The study of the relationship between phenology, plant community, and endophytic fungi provides a comprehensive view of the dynamics and health of ecosystems, with a specific focus on *Malesherbia auristipulata* [24]. The results of this research generate new hypotheses about how the phenology of endemic plants might influence the availability of resources for the surrounding flora and endophytic fungi [16,25]. For example, the flowering and fruiting period of *M. auristipulata* could affect the availability of pollen and nectar for pollinators, which in turn could impact the distribution and phenology of other plants in the community [26,27]. Additionally, the plant community in the same habitat as *M. auristipulata* could influence its phenology and vice versa through competition for resources such as light, water, and nutrients [28]. These types of plant–plant and plant–fungi interactions are fundamental for understanding the ecology of arid ecosystems and can have significant implications for biodiversity conservation and the management of these unique habitats [29].

Another hypothesis suggests that endophytic fungi can have significant effects on the phenology and health of host plants. Some of these fungi may promote plant growth and resistance to diseases, while others may act as pathogens or have neutral effects [19,30]. Therefore, the presence of certain endophytic fungi in *Malesherbia auristipulata* could influence its phenology and its interactions with the plant community [17,29].

It is important to consider that extreme events can have both short-term and long-term effects on phenology, the composition of the plant community, and the diversity of endophytic fungi in the ecosystem. These events can alter interactions between plants, and between plants and fungi, affecting the health and dynamics of *Malesherbia auristipulata* populations and other species in the region [31,32]. Such changes could significantly modify the structure and function of the ecosystem, highlighting the importance of understanding ecological interactions and the resilience of species in the face of extreme climatic events [29].

This study highlights the complexity of the relationships between phenology, plant community, and endophytic fungi and how these interactions can influence the structure and function of the ecosystems where *Malesherbia auristipulata* is found. A detailed analysis of these interactions provides valuable information for the conservation and management of this ecosystem [33,34]. Furthermore, it emphasizes the importance of understanding the dynamics of microbes, both in natural environments and in their relationship with host organisms, which enhances our appreciation of microbial diversity and its crucial role in terrestrial ecosystems [35,36].

Microbes are ubiquitous on Earth and possess the ability to survive in a wide variety of environments, ranging from hydrothermal vents in the deep ocean to deserts and polar regions [34]. Their ability to colonize even plant trichomes underscores the remarkable adaptability and diversity of microbes in different ecological niches [37].

Trichomes originate from epidermal cells and develop on the surface of various plant organs. These trichomes not only serve physical functions, such as protecting the plant against herbivores and increasing its stress tolerance (Figure 1D) [38,39,40], but they can also serve as habitats and infection sites for a variety of microbes, including saprophytic and pathogenic fungi [15].

The associations between fungi and plant trichomes have been the subject of a growing number of studies, which have revealed that trichomes can facilitate the adhesion and growth of fungi on the plant surface, creating a microenvironment conducive to their development [41]. These interactions can have significant implications for plant health and disease dynamics. An example of this is Tea Anthracnose, where the pathogenic fungus infects the plant through foliar trichomes, causing the characteristic leaf spots [42].

The case of *Malesherbia auristipulata* compared to other species, such as *Pinus pinaster*, shows how the relationship between the plant and the fungus *Fusarium circinatum* can vary significantly, from symbiosis to a pathogenic interaction. In *M. auristipulata*, *F. circinatum* appears to behave symbiotically without causing visible damage to the plant, suggesting that both species have developed an adaptive balance in the extreme environment of the Atacama Desert. However, in introduced species such as *P. pinaster*, *F. circinatum* acts as an aggressive pathogen, responsible for pitch canker, a serious disease affecting forest plantations [43]. In these pines, starting from the fourth day post-inoculation, the fungus begins to colonize the tissues, triggering a series of phytohormonal responses, including alterations in the levels of jasmonic acid (JA), salicylic acid (SA), ethylene (ET), and auxins, which facilitate the invasion of the pathogen [44,45]. These hormonal changes inhibit the plant’s natural defenses, allowing the fungus to spread and effectively damage the host tissue [45].

The study of the diversity of endophytic fungi associated with *Malesherbia auristipulata* is based on their genetic identity, morphological characteristics, and their presence in different plant organs such as the root, stem, and leaf. Recent research, including the morpho-molecular characterization of endophytic fungi from traditional medicinal plants like *Adhatoda vasica* and *Cassia alata*, highlights the effectiveness of ITS sequencing for identifying fungal species and understanding their ecological roles [46]. These fungi play a significant role in the plant’s adaptation to the extreme conditions of the Atacama Desert. By improving water retention, nutrient absorption, and plant defense mechanisms, endophytic fungi are crucial for survival in harsh environments [15,16].

In addition to facilitating plant growth and tolerance to drought, endophytic fungi also contribute to broader ecological interactions within plant communities. Studies have shown that fungi such as *Alternaria* and *Fusarium*, which were found in *M. auristipulata*, play important ecological roles in mitigating the effects of pathogens [17]. This symbiotic relationship helps explain the persistence of *M. auristipulata* in an ecosystem where the availability of water and nutrients is limited [27].

Furthermore, recent studies on the functional roles of these endophytic fungi provide a clearer understanding of their contributions to plant health and ecosystem stability. For example, *Fusarium circinatum* and *Alternaria* species found in this study have been shown to play roles in enhancing plant growth and protecting against biotic stressors [30]. The presence of these fungi in various plant organs suggests that they may have organ-specific functions, with some focusing on nutrient uptake from the roots, while others aid in defense in the stems and leaves.

Recent research on *Amaranthus tricolor* has demonstrated that different plant phenotypes can recruit distinct microbial communities in the rhizosphere, suggesting a strong influence of plant traits on microbial assembly [47]. This parallels our findings in *Malesherbia auristipulata*, where specific fungal taxa may associate more closely with particular phenological stages, enhancing the plant’s ability to adapt to extreme conditions. Additionally, recent studies have demonstrated the antagonistic activity of endophytic fungi isolated from medicinal plants such as *Ocimum basilicum* and *Leucas aspera*, which inhibit the growth of phytopathogens like *Fusarium oxysporum* and *Aspergillus niger* [48]. These findings align with the protective role that endophytic fungi may play in *M. auristipulata*, helping to maintain plant health and defend against pathogens, which is essential for survival in extreme environments like the Atacama Desert.

## 3. Materials and Methods

Geographic Location. This study was conducted at Cuesta El Águila, located in the Arica and Parinacota Region of Chile. This site was selected because it is the only known natural habitat where *Malesherbia auristipulata* forms communities alongside other associated flora species. Cuesta El Águila is part of Quebrada Cardones in the Sierra de Huaylillas (coordinates: S18°28′547″ and W69°51′485″).

Flora Associated with *Malesherbia auristipulata*. To characterize the plant community of *M. auristipulata*, samples were collected from all herbaceous and shrub species growing in association with this plant at Cuesta El Águila. A 1000 m transect was surveyed in the lower part of the quebrada, selecting vegetative and reproductive parts of all plants within a 10 m radius around the *M. auristipulata* specimens under study. All plants were preserved as herbarium specimens and were compared with type species at the National Museum of Natural History for taxonomic identification.

Phenological Study. The monthly and annual phenological study of *Malesherbia auristipulata* began on 28 January 2019 by tagging 165 specimens, corresponding to the majority of adult plants from the only described community at Cuesta El Águila. Fifteen days later, in February 2019, a strong flash flood severely affected the area, sweeping away 90% of the tagged plants. Due to the population decrease, the study was restarted on 5 March 2019, and 38 adult plants were tagged, including 11 dormant plants, which allowed for the detection of phenophase changes over the course of a year. To increase the sample size, 23 seedlings were included, originating from the germination of seeds forming banks at the base of adult plants (Figure 4). The monthly expeditions allowed for detailed observations of the plants, considering growth, flowering, fruiting, and other phenological events, as well as visits from potential or effective pollinators. A plant was considered to be in flowering or fruiting only if 20% or more of the plant displayed the corresponding phenophase.

Sample Collection for the Study of Endophytic Mycoflora. Samples of seeds, roots, stems, and leaves of *M. auristipulata* were systematically collected at different times of the year to capture phenological changes and seasonal variability. During each expedition, samples were taken from various organs of different specimens. Only healthy, disease-free plants were selected to ensure a suitable environment for the study of endophytic microorganisms.

The plant samples were washed and disinfected following a strict protocol: first, they were immersed in 95% alcohol for 2 min, followed by 2% sodium hypochlorite for another 2 min, and finally in 70% alcohol for 2 min. Subsequently, the samples were rinsed with sterile distilled water. From this point onward, all procedures were carried out in a laminar flow hood to maintain sterility. The stem samples were cut longitudinally and placed in Petri dishes with specific culture media, such as PDA (potato dextrose agar) and WA (water agar). In the case of roots, the ends and edges were removed, and a piece from the inner zone was placed in the same culture media. The flower and leaf samples were macerated in a porcelain mortar, and the resulting pieces were placed in Petri dishes with PDA (potato dextrose agar) and WA.

Morphological and Molecular Characterization. The endophytic fungi isolated from *M. auristipulata* were studied both morphologically and molecularly. The macroscopic and microscopic characteristics of the fungal structures were described, and each isolate was cultivated in pure, monospore cultures, recording the macroscopic characteristics of the colonies and the microscopic elements of taxonomic value. From pure cultures, molecular analysis was performed for the precise identification of fungal species.

For this purpose, DNA was extracted from the isolated fungi, and subsequently the internal transcribed spacer region (*ITS*) was amplified using primers *ITS4* (5′-*TCCTCCGCTTATTGATATGC*-3); and *ITS5* (5′-*GGAAGTAAAAGTCGTAACAAGG*-3′) [49]. The obtained sequences were compared with available databases to determine the taxonomic identity of the species based on sequence similarity.

This combined approach of morphological and molecular characterization allowed for a robust identification of the endophytic fungal species present in *Malesherbia auristipulata*, providing a more comprehensive understanding of their diversity and potential function in the host plant.

## 4. Conclusions

The analysis of the plant community of *Malesherbia auristipulata* in its natural habitat has revealed a diverse plant community well adapted to the extreme conditions of the Atacama Desert. The botanical species that make up this community are native and endemic, with no evidence of introduced species, which is consistent with previous studies on the vegetation of this region [5,20]. The high proportion of endemic species underscores both the uniqueness and fragility of this ecosystem, highlighting the urgent need for its conservation.

The phenology of *M. auristipulata* and its surrounding plant species provides essential information for understanding seasonal dynamics and ecological interactions in these arid habitats. Phenological records show how extreme climatic events can have drastic effects on the phenology and survival of plants, which in turn impacts the structure and function of the ecosystem [24,29].

The study of endophytic fungi associated with *M. auristipulata* has revealed a diverse fungal community with significant implications for the plant’s health and adaptation to its environment. The variability in the distribution of these fungi among different parts of the plant suggests complex interactions that may influence the phenology and resilience of *M. auristipulata* in the face of changing environmental conditions. The presence of endophytic fungi such as *Alternaria* and *Fusarium* underscores the importance of these microorganisms in the dynamics of plants in arid ecosystems [15,34].

The study of interactions between phenology, plant community, endophytic fungi, and plant trichomes offers a more comprehensive understanding of the dynamics and health of ecosystems. These interactions not only influence the structure and function of the ecosystem but also the health and performance of the host plants [15,16]. Research in this field is essential for the conservation and sustainable management of natural ecosystems, especially in arid regions where biodiversity and adaptation to extreme conditions are crucial [17,34].

Understanding the dynamics of microorganisms in natural environments and their relationship with host organisms is essential for appreciating microbial diversity and its role in terrestrial ecosystems [35,36]. Microbes, capable of colonizing a wide range of environments—from deserts to poles—demonstrate an adaptability and diversity that underscores their importance in various ecological niches [39], especially in arid ecosystems such as the Atacama Desert [20].

## Figures and Tables

**Figure 1 plants-13-03035-f001:**
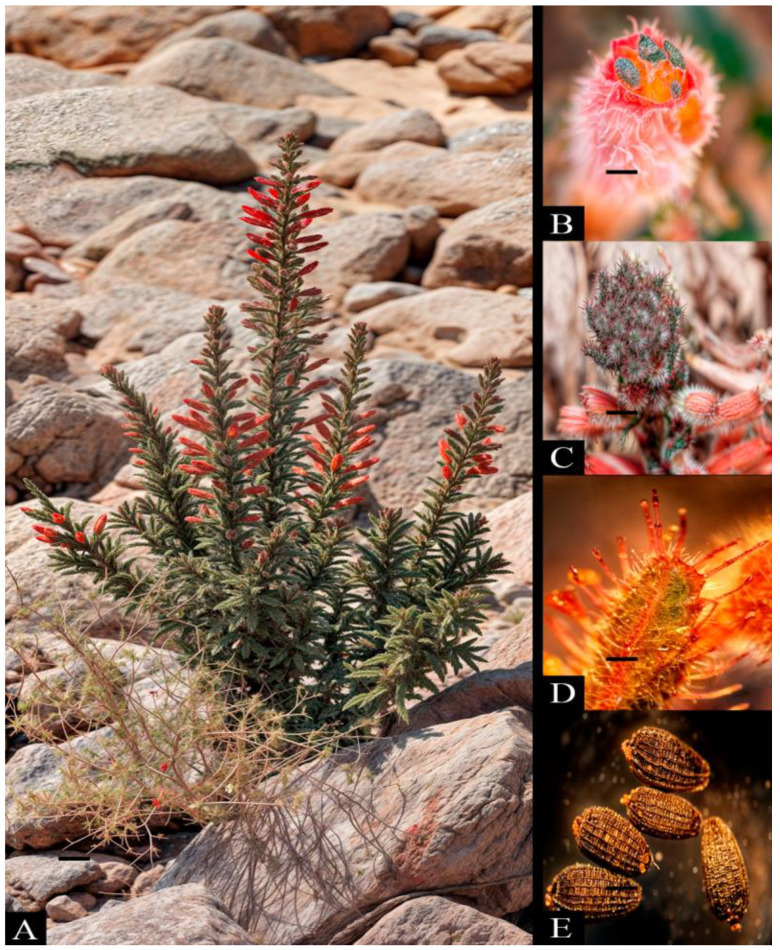
*Malesherbia auristipulata* in its natural habitat, Cuesta El Águila, Quebrada Cardones, Arica and Parinacota Region: (**A**) adult plant, (**B**) flower, (**C**) vegetative buds, (**D**) glandular trichomes on leaves, and (**E**) seeds.

**Figure 2 plants-13-03035-f002:**
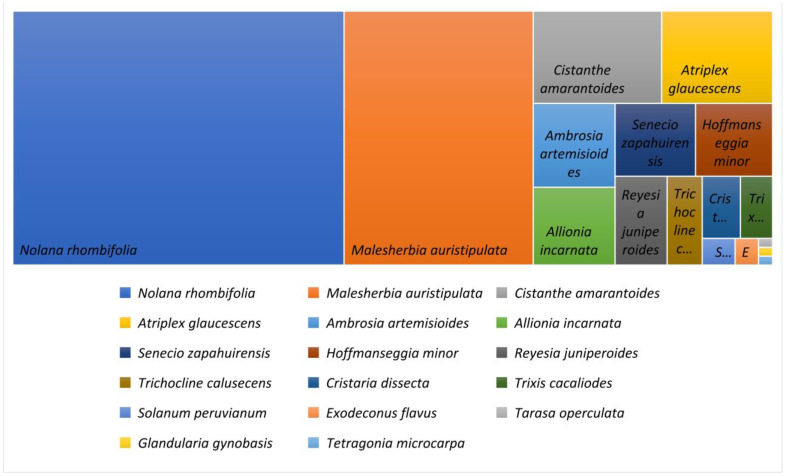
Abundance (number of individuals) recorded at 49 sampling points along a 20 m transect, Cuesta El Águila, Quebrada de Cardones, Arica and Parinacota Region.

**Figure 3 plants-13-03035-f003:**
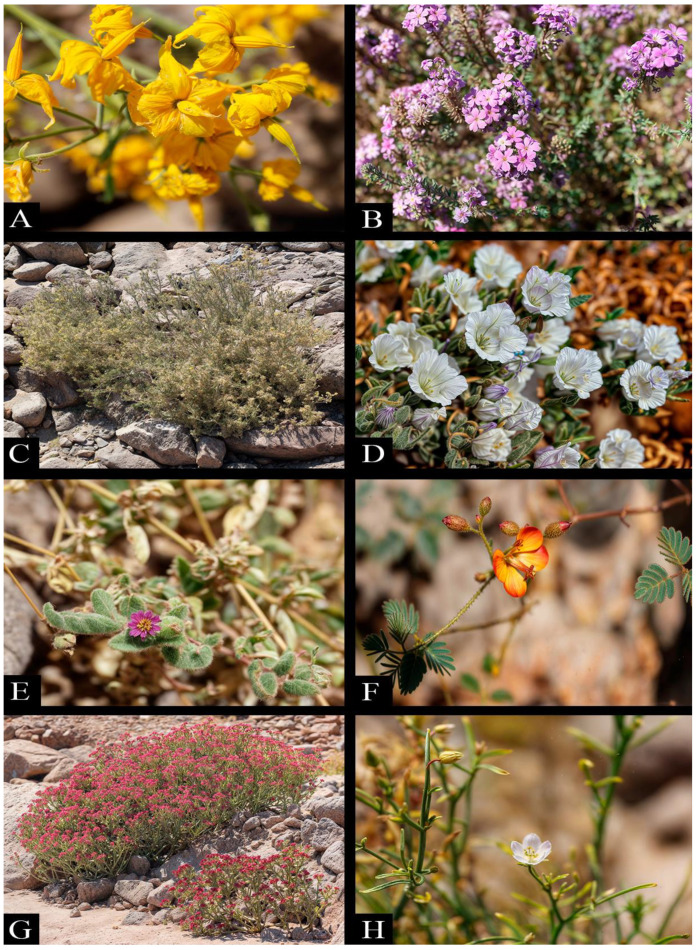
Plant species that are part of the plant community of *Malesherbia auristipulata*: (**A**) *Solanum peruvianum*, (**B**) *Glandularia gynobasis*, (**C**) *Senecio zapahuirensis*, (**D**) *Nolana rhombifolia*, (**E**) *Allionia incarnata*, (**F**) *Hoffmannseggia minor*, (**G**) *Cistanthe* sp., and (**H**) *Spergularia fasciculata*.

**Figure 4 plants-13-03035-f004:**
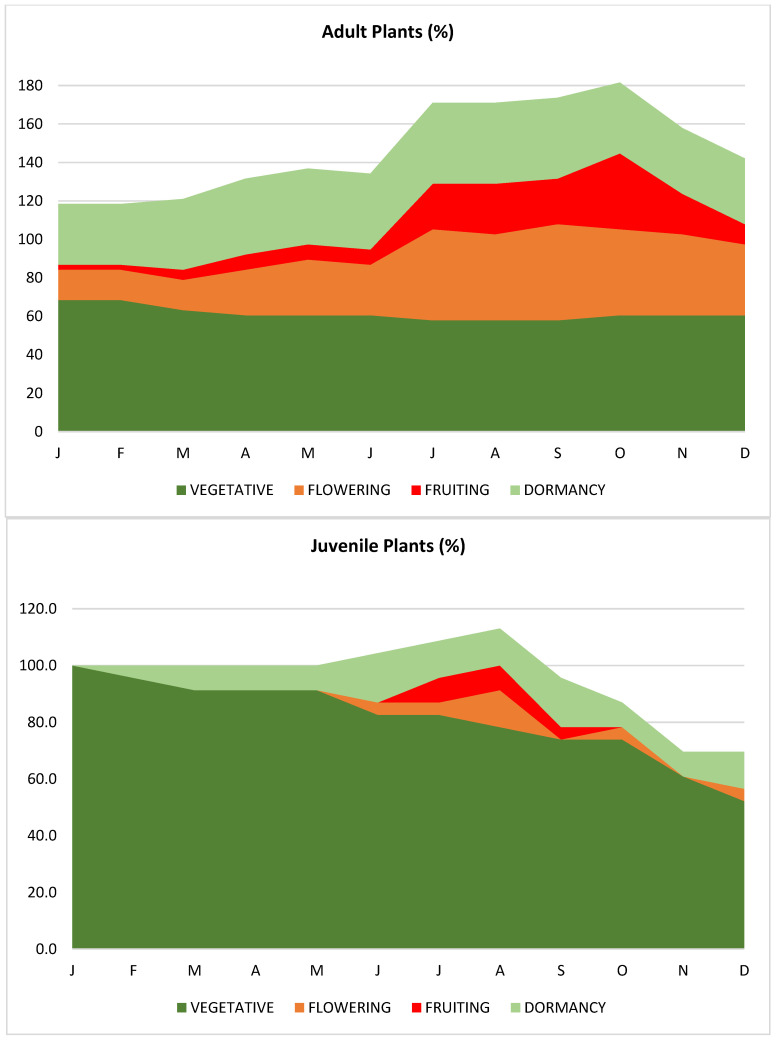
Stacked area charts representing the phenological behavior of *Malesherbia auristipulata* throughout the year, comparing the growth phases between juvenile and adult plants in percentage terms. The charts are divided into four phenological categories: vegetative, flowering, fruiting, and dormancy.

**Figure 5 plants-13-03035-f005:**
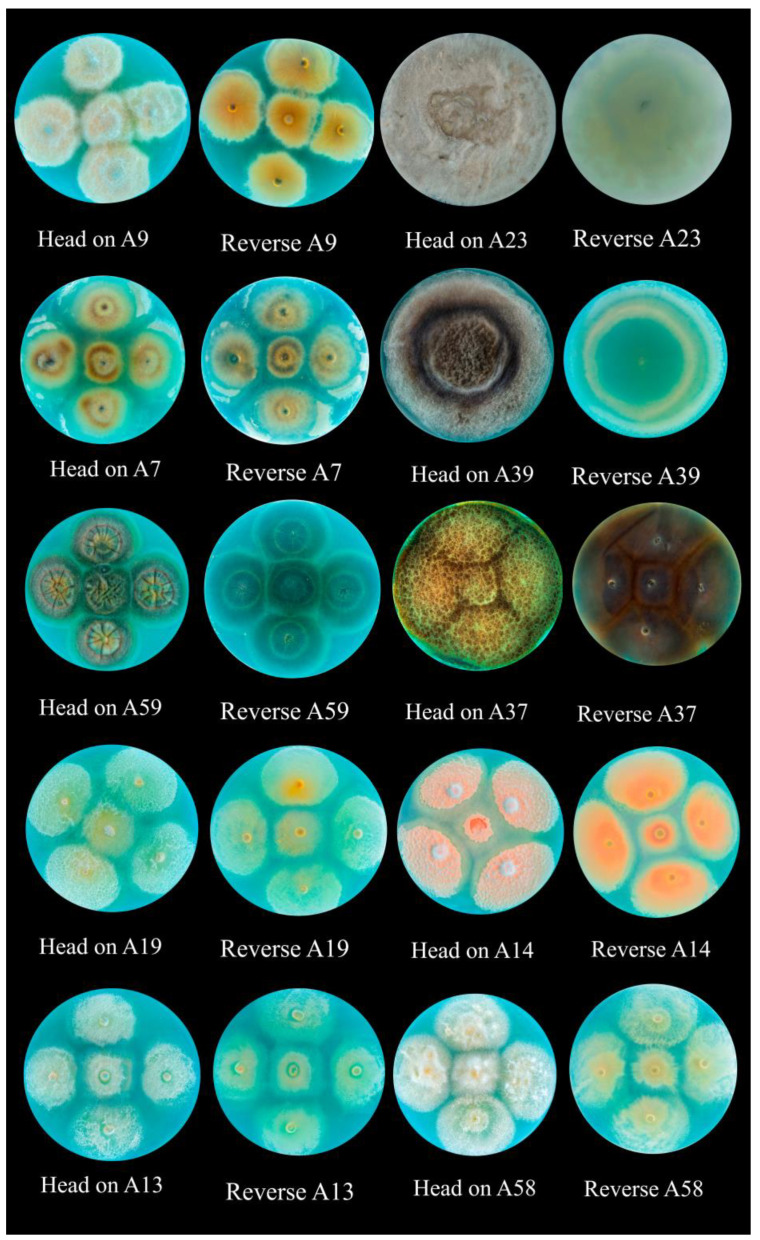
Colonies of endophytic fungi on APD medium, isolated from *Malesherbia auristipulata*. A9: *Alternaria alstroemeriae* NR_163686.1, A23: *A. alstroemeriae* NR_163686.1, A7: *A. sorghi* NR_160246.1, A39: *A. sorghi* NR_160246.1, A59: *Aureobasidium melanogenum* NR_159598.1, A37: *Didymella keratinophila* NR_158275.1, A19: *Fusarium biomiforme* NR_111885.1, A14: *F. circinatum* NR_120263.1, A13: *F. nurragi* NR_159860.1, and A58: *F. nurragi* NR_159860.1.

**Table 1 plants-13-03035-t001:** Plant community of *Malesherbia auristipulata* in Cuesta El Águila, Quebrada Cardones, Arica and Parinacota Region, Chile.

Family	Scientific Name	Common Name	Native/Endemic	Habit	Distribution *
*Aizoaceae*	*Tetragonia microcarpa* Phil.	Aguanosa	native	annual herb	AYP, TAR, ANT, ATA
*Asteraceae*	*Ambrosia artemisioides* Meyen & Walp. ex Meyen	Tikara, Pikara, Cadillo, Chaspaksa, Monte verde, Tola negra, Pegapega, Lipelipe	native	shrub	AYP, TAR, ANT
*Asteraceae*	*Senecio zapahuirensis* Martic. & Quezada	Desconocido	endemic	shrub	AYP, TAR
*Asteraceae*	*Trichocline caulescens* Phil.	Wanti, Garra de león, Bailabaila	endemic	perennial herb	AYP, TAR, ANT
*Asteraceae*	*Trixis cacalioides* (Kunth) D. Don	Visavisa	native	shrub	AYP, TAR, ANT, ATA
*Chenopodiaceae*	*Atriplex glaucescens* Phil.	Juirajuira, Piyaya	endemic	shrub	AYP, TAR, ANT, ATA
*Chenopodiaceae*	*Chenopodium petiolare* Kunth.	Juirajuira, Kañawa, Quinua de gentiles, Piyaya hembra	native	perennial herb	AYP, TAR, ANT, ATA, COQ, VAL
*Fabaceae*	*Hoffmannseggia minor* (Phil.) Ulibarri	Algarrobilla, Bilankichu, Kulchau, Mutukuru, Motokoro	native	perennial herb	AYP, TAR, ANT
*Malvaceae*	*Cristaria dissecta* Hook. & Arn. var. Dissecta	Malvavisco, Malva	native	perennial herb	AYP, TAR, ANT, ATA, COQ, VAL, RME, LBO
*Malvaceae*	*Tarasa operculata* (Cav.) Krapov.	Poq’ot’ola, Qhella hembra, Qhella blanca, Malva, Tarasa	native	shrub	AYP, TAR, ANT
*Montiaceae*	*Cistanthe amarantoides* (Phil.) Carolin ex Herschkovitz	Anojarjinchu, Tiqintiqi, Oreja de chancho	endemic	perennial herb	AYP, TAR, ANT, ATA, COQ
*Nycataginaceae*	*Allionia incarnata* L. (hierba perenne, nativa)	Desconocido	native	perennial herb	AYP, TAR, ANT, ATA
*Passifloraceae*	*Malesherbia auristipulata* Ricardi	Ají de zorra, Piojillo	endemic	shrub	AYP, TAR
*Solanaceae*	*Exodeconus flavus* (I.M. Johnst.) Axelius & D’Arcy	Desconocido	native	annual herb	AYP, TAR
*Solanaceae*	*Nolana rhombifolia* Martic. & Quezada	Suspiro	endemic	annual herb	AYP
*Solanaceae*	*Reyesia juniperoides* (Werderm.) D’Arcy	Canchalahua	endemic	perennial herb	AYP, TAR
*Solanaceae*	*Solanum peruvianum* L.	Tomatillo	native	perennial herb	AYP, TAR
*Verbenaceae*	*Glandularia gynobasis* (Wedd.) N. O’Leary & P. Peralta	Nametusangaya, Mamapasankayo, Flor del campo	endemic	perennial herb	AYP, TAR

*: Distribution in Chile: The abbreviations used in the table correspond to the country’s political–administrative divisions: AYP (Arica and Parinacota), TAR (Tarapacá), ANT (Antofagasta), ATA (Atacama), COQ (Coquimbo), VAL (Valparaíso), RME (Metropolitan Region of Santiago), and LBO (Libertador General Bernardo O’Higgins).

**Table 2 plants-13-03035-t002:** Annual phenology of adult and juvenile *Malesherbia auristipulata* plants in Cuesta El Águila, Quebrada Cardones, Arica and Parinacota Region. The numbers represent the percentage of individuals in each phenophase.

	Vegetative (%)		Flowering (%)		Fruiting (%)		Dead (%)	
Month/Phenophase	Adult	Juvenile	Adult	Juvenile	Adult	Juvenile	Adult	Juvenile
Jan.	81.6	100	15.8	0	2.6	0	0	0
Feb.	81.6	100	15.8	0	2.6	0	0	0
Mar.	73.7	100	21.1	0	5.3	0	0	0
Apr.	68.4	100	23.7	0	7.9	0	0	0
May	63.2	100	28.9	0	7.9	0	0	0
Jun.	65.8	95.7	26.3	4.3	7.9	0	0	0
Jul.	57.9	82.6	28.9	4.3	13.2	8.7	0	4.3
Aug.	57.9	73.9	26.3	8.7	15.8	8.7	0	8.7
Sep.	57.9	86.9	36.8	0	5.3	4.4	0	8.7
Oct.	61.1	78.3	25.0	4.3	13.9	0	1	17.4
Nov.	61.8	69.6	17.6	0	20.6	0	2	30.4
Dec.	62.9	56.5	17.1	4.3	20	0	2	39.2

**Table 3 plants-13-03035-t003:** Diversity of endophytic fungi associated with *Malesherbia auristipulata*.

Code	BLASTN Identity	% Identity	Root	Stem	Leaf
*A9*	*Alternaria alstroemeriae NR_163686.1*	*99.82*	+		
*A23*	*Alternaria alstroemeriae NR_163686.1*	*100*	+		
*A7*	*Alternaria sorghi NR_160246.1*	*99.83*	+		
*A17*	*Alternaria sorghi NR_160246.1*	*99.83*		+	
*A30*	*Alternaria sorghi NR_160246.1*	*99.65*	+		
*A34*	*Alternaria sorghi NR_160246.1*	*99.13*		+	
*A36*	*Alternaria sorghi NR_160246.1*	*99.65*		+	
*A39*	*Alternaria sorghi NR_160246.1*	*99.65*		+	
*A44*	*Alternaria sorghi NR_160246.1*	*99.83*	+		
*A55*	*Alternaria sorghi NR_160246.1*	*99.65*		+	
*A61*	*Alternaria sorghi NR_160246.1*	*99.65*		+	
*A62*	*Alternaria sorghi NR_160246.1*	*99.83*		+	
*A63*	*Alternaria sorghi NR_160246.1*	*99.83*		+	
*A59*	*Aureobasidium melanogenum NR_159598.1*	*99.49*		+	
*A37*	*Didymella keratinophila NR_158275.1*	*99.82*		+	
*A19*	*Fusarium biomiforme NR_111885.1*	*84.83*	+		
*A14*	*Fusarium circinatum NR_120263.1*	*97.85*			+
*A13*	*Fusarium nurragi NR_159860.1*	*92.93*		+	
*A58*	*Fusarium nurragi NR_159860.1*	*91.96*	+		

## Data Availability

Data are contained within the article.

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
