# Peer review of "Phenological and Fungal Interactions of Malesherbia auristipulata Ricardi (Passifloraceae) in the Atacama Desert: Adaptations and Conservation in an Extreme Ecosystem"

_plants, 2024, doi:10.3390/plants13213035_

Round 1
Reviewer 1 Report
Comments and Suggestions for Authors
The article is well written and needs to be improved with the following minor changes. The paper represent the novel idea and methodology and the idea is clear.
1. The methodology section could benefit from a more detailed explanation of the sampling techniques and statistical analyses used. This would enhance reproducibility and allow readers to better understand the robustness of the findings.
2. Consider improving the clarity of tables and figures by including legends that explain the significance of the data presented. For instance, Table 1 could include a brief description of why certain species were chosen for comparison, enhancing its interpretative value.
3.The discussion section should explicitly connect the findings on endophytic fungi to broader ecological implications. This could involve discussing how these fungi contribute to plant resilience in extreme environments, thereby linking your results to existing literature more effectively.
4. Integrate more recent studies on endophytic fungi and their roles in plant ecology to provide a contemporary context for your findings. This would strengthen the article's relevance and demonstrate how it fits into ongoing research in the field.
Comments on the Quality of English LanguageThe English is fine
Author Response
- The methodology section could benefit from a more detailed explanation of the sampling techniques and statistical analyses used. This would enhance reproducibility and allow readers to better understand the robustness of the findings.
The paragraph corresponding to line 358 was enriched with methodological details referring to the sample collection process and included elements that defined the number of samples and the spectrum of individuals studied. In addition, the phenological study procedure was detailed.
2. Consider improving the clarity of tables and figures by including legends that explain the significance of the data presented. For instance, Table 1 could include a brief description of why certain species were chosen for comparison, enhancing its interpretative value.
Indeed, the reference of the tables was improved. The plant species studied correspond to the total vegetation community that accompanies Malesherbia. There was no discrimination in including one species or another, the total was considered. This is indicated in the text in its new version.
3.The discussion section should explicitly connect the findings on endophytic fungi to broader ecological implications. This could involve discussing how these fungi contribute to plant resilience in extreme environments, thereby linking your results to existing literature more effectively.
We present Results and Discussion. For this reason, a figure (Figure 2) and an explanation were included in line 123, which enriches the discussion and reinforces the data presented. On line 178, 2 figures were added that represent the phenological evolution of plant species in a year of records, both for adult and juvenile specimens. On li plants-3226393
ne 186 a paragraph is added that deepens the discussion. In the same way, in line 326 the paragraph is reinforced with new and current perspectives that seek to connect the phenological development of the plant with the presence of endophytic fungi.
Yes, current references were included and they correspond to reference works, so we believe that they respond to the evaluator's observation. We focus attention on Malesherbia and not on another species of the vegetation community due to the important adaptive, unique and relevant characteristics it presents. In addition, there is specific information on the presence of phytochemicals in its leaves, trichomes and other organs, which supports the relevance of knowing in detail this plant species and some of its microbial "partners."
4. Integrate more recent studies on endophytic fungi and their roles in plant ecology to provide a contemporary context for your findings. This would strengthen the article's relevance and demonstrate how it fits into ongoing research in the field.
Does the introduction provide sufficient background and include all relevant references?
Updated information on endophytic fungi and their functions in plant ecology was included (for example, Akram, S.; Ahmed, A.; He, P.; He, P.; Liu, Y.; Wu, Y.; Munir, S.; He, Y. Uniting the role of endophytic fungi against plant pathogens and their interaction. J. Fungi 2023, 9, 72-95; Lin, X.-R.; Yang, D.; Wei, Y.-F.; Ding, D.-C.; Ou, H.-P.; Yang, S.-D. Amaranth Plants with Various Color Phenotypes Recruit Different Soil Microorganisms in the Rhizosphere. Plants 2024, 13, 2200; Devi, Wairokpam & Kannaiah, Surendirakumar. In vitro antagonistic activity of endophytic fungi associated with medicinal plants of Lamiaceae towards phytopathogenic fungi. Journal of Mycopathological research 2024, 62. 447-451.

Reviewer 2 Report
Comments and Suggestions for Authors
Please find the attachment of the manuscript with minor corrections.
The manuscript "Phenological and Fungal Interactions of Malesherbia auristipulata Ricardi (Passifloraceae) in the Atacama Desert: Adaptations and Conservation in an Extreme Ecosystem" is extremely interesting and it would be interesting to Plants readers. However, a few issues have to be addressed.
First, in the Introduction, there is much information on the desert. However, more information is needed on Malesherbia auristipulata.
The research is appropriately designed, but the Materials and Methods section has to be improved with a detailed description of
- how many plants were evaluated (percentage is not enough in this case)
- how they were evaluated
- How and when was the plant material harvested and processed to perform molecular analysis of endophytic fungi? What primers were used in this study?

Minor editing of English language is required.
Author Response
- First, in the Introduction, there is much information on the desert. However, more information is needed on Malesherbia auristipulata.
Malesherbia auristipulata is a relict species, it occupies a very restricted space in the Atacama Desert and there is little information available. We reference all the available information plus our own observations
2. The research is appropriately designed, but the Materials and Methods section has to be improved with a detailed description of
- how many plants were evaluated (percentage is not enough in this case)
Because it is a plant species at risk, our methodology did not consider extracting plant material from the site, except what was necessary to study endophytic fungi. In addition, the entire place where the plants grow was studied and 90% of the specimens that were present were evaluated. Figures (graphs) were included in the MS and the evaluation method on the plant population is detailed. In addition, we integrate all plant species under the plant community concept.
3. - How and when was the plant material harvested and processed to perform molecular analysis of endophytic fungi? What primers were used in this study?
The collection of plant material for the study of endophytic fungi was limited due to the intrinsic value of the species. The samples were taken monthly, in each expedition, which is indicated in the MS.
DNA was extracted from the isolated fungi, and subsequently the internal transcribed spacer region (ITS) was amplified using primers ITS4 (5'-TCCTCCGCTTATTGATATGC-3); and ITS5 (5'- GGAAGTAAAAGTCGTAACAAGG-3 ') (White et al., 1990). This information was included in the MS
